Comparison of fracture resistance between immediate and delayed composite restorations with or without fiber after root canal treatment: a field-emission-gun scanning electron microscope study

Kaynar Zeynep Buket 1 buket_karakus@hotmail.com
Akbal Dinçer Gözde 1
Donmez Nazmiye 2
1 Faculty of Dentistry, Istanbul Okan University , Istanbul , Turkey
2 Faculty of Dentistry, Abant Izzet Baysal University , Bolu , Turkey
Abu Hasna Amjad
Electronic publication date: 2025 Feb 25
Publication date: 2025
Volume: 13
Electronic Location ID: e19018
Received 2024 Jul 3; Accepted 2025 Jan 28
Copyright: © 2025 Kaynar et al.
Copyright year: 2025
Copyright holder: Kaynar et al.
License: This is an open access article distributed under the terms of the Creative Commons Attribution License, which permits unrestricted use, distribution, reproduction and adaptation in any medium and for any purpose provided that it is properly attributed. For attribution, the original author(s), title, publication source (PeerJ) and either DOI or URL of the article must be cited.
License URL: https://creativecommons.org/licenses/by/4.0/

Keywords: Composite, Delayed restorations, Root canal treatment, Polyethylene fiber, Fe-sem

Funding: The authors received no funding for this work.

==============================
Background

This study aimed to evaluate the fracture resistance of immediate and delayed restorations after root canal treatment (RCT).

Methods

Sixty human intact premolar teeth were used. Teeth were prepared disto-occlusally. After RCT, teeth were randomly divided into two groups according to restoration times (immediate or 3 months delayed). The three month delay was imitated by thermomechanical aging. Then, samples were divided into six subgroups (n = 10). While I (Composite resin), II (Flowable bulk-fill) and III (Ribbond+Flowable bulk-fill) were restored immediately, Groups IV, V and VI were restored with temporary filling material and stored in distilled. After subjected to thermo-cycling (2,500 cycles, 5–55 °C) and exposed to 60,000 cycles in a chewing simulator, Group IV (Composite resin), V (Flowable bulk-fill) and VI (Ribbond+Flowable bulk-fill) were restored. All of the teeth were fractured on the universal testing machine. Failure modes were analyzed using scanning electron microscope. Data were analyzed using the Shapiro-Wilk and two-way ANOVA tests (p < 0.05).

Results

The highest fracture resistance was recorded in Group III while the lowest in Group VI. No statistically significant difference was observed among groups (p > 0.05). Most of the repairable fractures were seen in Groups I and II.

Conclusion

Delaying the permanent restorations of teeth for 3 months did not affect fracture resistance. However, it was suggested to avoid delaying restorations for obtaining repairable surfaces.

Introduction

The long-term success of root canal treatment (RCT) is related to preparation and irrigation procedures, obturation techniques, and the quality of the restoration of endodontically treated teeth (ETT) (Kirkevang et al., 2000; Tronstad et al., 2000; Gillen et al., 2011).

Clinicians are worried about the fracture risk of teeth after RCT. The remaining enamel and dentin tissue is one of the most important factors in determining the fracture resistance of the teeth (Oliveira, Denehy & Boyer, 1987). Large cavities and weak cavity walls are frequently observed after RCT, especially in posterior teeth (Yasa et al., 2016). During the cavity preparation, the removal of the marginal ridge has a negative effect on the fracture resistance of ETT (González-López et al., 2006). In addition, it has been reported that dehydrated dentin and loss of collagen cross-links after RCT reduce the fracture resistance of the teeth (Oskoee et al., 2009).

The restoration of the ETT is a challenge. Recently, new restorative materials were developed to strengthen the mechanical properties of weak teeth (Chrepa et al., 2014). Although indirect restorations are considered the optimal choice for ETT, improved composite resin materials are also one of the best alternatives for the indirect restoration of ETT (Kemaloglu et al., 2015). Direct composite resin restorations (CRR) have some advantages, such as being repairable, fast, affordable, esthetic, and conservative, but it is still unclear whether CRR has enough fracture resistance for ETT (Fokkinga et al., 2005; Kemaloglu et al., 2015).

Polymerization shrinkage is another cause of the failure of CRR (Garapati et al., 2014; Benetti et al., 2015). An incremental technique is applied to reduce polymerization shrinkage. In the case of extensive caries, especially after RCT, the staged technique is time-consuming and requires technical precision for dentists (Garapati et al., 2014). Therefore, bulk-fill technology is introduced to be placed in a thickness of 4–5 mm and single-cured (El-Damanhoury & Platt, 2014). In addition to the time savings of this application, it was reported that bulk-fill composites show lower polymerization stress than conventional composites (El-Damanhoury & Platt, 2014). Bulk-fill composites have two types; flowable and paste. Since flowable bulk-fill composites showed weaker mechanical properties than paste types, application areas of flowable bulk-fill composites were limited with narrow cavities (Parra Gatica, Duran Ojeda & Wendler, 2023). With the development of the content and the amount of filler in flowable bulk-fill composites, manufacturers claim that flowable bulk-fill composites can be used safely in the same cavities as paste-type composites (Parra Gatica, Duran Ojeda & Wendler, 2023). Considering the ease of application, flowable bulk-fill composite was used in this study.

Another technique for restoring ETT is the use of polyethylene fibers. Polyethylene fibers reinforce the mechanical properties of CRR (Lassila, Nohrström & Vallittu, 2002; Kemaloglu et al., 2015). Polyethylene fibers have high elasticity and low flexural modulus (Lassila, Nohrström & Vallittu, 2002). Due to these properties, polyethylene fibers prevent crack formation by dispersing and reducing stresses (Shafiei et al., 2014).

It was pointed out that applying post-endodontic restorations immediately following RCTs was better (Mindiola et al., 2006). However, restoration procedures can be delayed due to patient—or clinician-related reasons. Dental treatment can also be interrupted in unexpected situations, such as the COVID-19 pandemic.

Therefore, the purpose of this study was to evaluate and compare the fracture resistance of the immediately applied and delayed restorations in premolar teeth after RCT. The null hypotheses of this study are: 1) Fiber insertion under the flowable bulk-fill composite does not affect the fracture resistance of ETT. 2) Delayed restoration will not reduce the fracture resistance of ETT.

Materials and Methods

This study was approved by the Bezmialem Vakif University Ethics Committee (E-54022451-050.05.04-89852). Following a power calculation based on a 25% difference in the prevalence of fracture resistance between all groups (power of 0.8 and 5%), the required sample size was calculated as 10 teeth per group. Accordingly, sixty maxillary premolar teeth with similar sizes, mature and extracted for orthodontic reasons were included in this study. The teeth with any cracks or fractures were excluded. Tooth extraction was performed after the participants signed an informed consent form. After extraction, debris and soft tissue remnants were cleaned, and the teeth were kept in distilled water.

Endodontic treatment procedures

Standardized disto-occlusal (DO) cavities (buccolingual width = 4 mm, mesiodistal width = 2.5, 1 mm above cementoenamel junction) were prepared (Kucukyilmaz et al., 2015). After preparing the endodontic access cavity, the pulp was extirpated. Then, the working length was determined as being 1 mm short of the apical foramen by inserting a #10 K-file (Dentsply Maillefer, Tulsa, OK, USA). The root canals were prepared with ProTaper Next Rotary instruments (Dentsply, Ballaigues, Switzerland) up to X3 file at the working length and irrigated with 2 ml %2.5 NaOCl (Microvem AF, Istanbul, Turkey) solution after each instrument. A final flush was applied using 5 ml of %2.5 NaOCl (Microvem AF, Istanbul, Turkey), and distilled water, respectively. Then the root canals were dried using paper points. Root canals were obturated with cold lateral condensation technique using AH plus (Dentsply DeTrey GmbH, Konstanz, Germany) and tapered gutta-perchas (Dentsply, Ballaigues, Switzerland).

After RCT, teeth were divided randomly into two groups according to the restoring time (immediate or 3 months delayed). Delayed teeth groups were temporarily restored with conventional glass ionomer cement (3M ESPE; Ketac Molar Easymix, Germany) until the thermocycling was completed. Then, these two groups were divided into three subgroups according to the restorative materials: nanohybrid composite, flowable bulk-fill composite, or polyethylene fiber under the flowable bulk-fill composite.

The experimental groups:

Selective etching was used for six groups. Enamel surfaces were etched with 37% ortho-phosphoric acid (Pulpdent Corp., Watertown, MA, USA) for 15 s, rinsed, and dried. Then, a universal self-etch adhesive system (Tokuyama Bond Force II, Tokuyama Dental, Tokyo, Japan) was applied to the cavities according to the manufacturer’s instructions (Tokuyama, 2018).

Immediate restorations after RCT

Group I: The cavities were restored with nano-hybrid composite resin (Tokuyama Estelite Posterior, Tokuyama Dental, Tokyo, Japan) immediately after RCT.

Group II: The cavities were restored with flowable bulk-fill composite resin (Tokuyama Estelite Bulk-fill, Tokuyama Dental, Tokyo, Japan) immediately after RCT.

Group III: Polyethylene fiber (Ribbond, Seattle, WA, USA) was cut and wetted with Tokuyama Bond Force.

Placement of Ribbond

Ribbond was stored in a dark box until the restoration procedures. Flowable bulk fill composite resin was applied to the pulpal, buccal, and lingual walls. Before curing, Ribbond was placed on the cavity floor to the 2/3 mm of the buccal and lingual wall (Eliguzeloglu Dalkılıç et al., 2019).

Then, cavities were restored with flowable bulk-fill composite resin. Figure 1 demonstrates the placement of the Ribbond in the cavity.

Figure 1 Demonstration of placement of Ribbond.

Storage and thermo-mechanical aging procedures

Restorations delayed for 3 months were simulated by thermomechanical-aging procedures. For the thermal cycling procedure, the teeth were subjected to 2,500 cycles (5–55 °C) with a dwell time of 30 s and a transfer interval of 10 s (SD Mechatronic, Feldkirchen-Westerham, Germany). Then, samples were submitted to mechanical loading (SD Mechatronic Chewing Simulator CS 4-2, Feldkirchen-Westerham, Germany) with 60,000 load cycles at a frequency of 1.7 Hz to a vertical load of 50 N (Beuer et al., 2012; Heydecke, Zhang & Razoog, 2001). During the mechanical loading test, the samples were submerged in distilled water.

After thermo-mechanical aging for 3 months:

The temporary restorations were removed conventionally with a diamond bur.

Group IV: The cavities were restored with nano-hybrid composite resin.

Group V: The cavities were restored with flowable bulk-fill composite resin.

Group VI: Ribbond was prepared and inserted as Group III, and cavities were restored with flowable bulk fill composite resin.

Each group’s samples were light-cured (3M Elipar Deepcure S, St. Paul, MN, USA) for 20 s.

Soflex discs (3M ESPE, St. Paul, MN, USA) were used for finishing and polishing. Then, all specimens were embedded in self-curing acrylic resin (Imicryl, Konya, Turkey).

The materials used in this study are described in Table 1.

Table 1 Materials used in the present study and their compositions.

Type of materials	Manufacturers	Lot number:	Compositions	
Tokuyama Bond Force II	Tokuyama Dental Corporation, Tokyo, Japan	3,271	Phosphoric acid monomer, Bis-GMA, TEGDMA, HEMA, Alcohol, Water, Camhorquinone	
Estelite Sigma Quick A2 nano-hybrid posterior composite resin	Tokuyama Dental Corporation, Tokyo, Japan	W1771	Organic matrix; Bis-GMA, TEGDMA, Bis-MPEPP	
Inorganic; Silica-zirconia	
Particle size: 2 um	
Particle size/ratio: 0.1–10 um	
Weight: 84% filler	
Volume: 70% filler	
Estelite Bulk-Fill Flow A2	Tokuyama Dental Corporation, Tokyo, Japan	0789	Bis-GMA, TEDGMA, Bis-MPEPP, Mequinol, Dibutyl hydroxyltoluene, UV absorber, spherical silica-zirconia filler, CQ, RAP, initiator system	
Ribbond (polyethylene fiber)	Ribbond Inc, Seattle, Washington, USA	+D758T0	Ultra-high molecular weight polyethylene, Homopolymer H-(CH2− CH2)n-H	

Fracture test and failure mode analysis using a field-emission-gun scanning electron microscope

After restorations were completed, specimens were fractured on the universal testing machine (Shimadzu, Tokyo, Japan). The compressive load was applied using a 3 mm stainless steel ball at crosshead speed 1 mm/min parallel to the long axis of the tooth until fracture occurred. The loads needed to fracture the samples were measured in Newtons (Eliguzeloglu Dalkılıç et al., 2019).

To evaluate failure surfaces, the samples were mounted on aluminum stubs and gold sputter coated using a gold-sputtering device (Ultra 55, Zeiss, Oberkochen, Germany). The fractured specimens were examined with a field-emission-gun scanning electron microscope (FE-SEM) (Apreo 2 SEM; Thermo Fisher Scientific, Waltham, MA, USA).

Failure modes were categorized as 1) Repairable; fracture line was at or above the CEJ; 2) Unrepairable; fracture line was more apical to the CEJ (Fig. 2) (Shafiei et al., 2014).

Figure 2 (A) Repairable fracture is seen in Group III. (B) SEM images of repairable fracture in Group III. (C) Unrepairable fracture is seen in Group VI. (D) SEM images of unrepairable fracture in Group VI. (E) Unrepairable fracture is seen in Group V. (F) SEM images of unrepairable fracture in Group V.

Statistical analysis

Data analyses were made by using IBM SPSS V22 (Chicago, IL, USA). The data were analyzed for normal distribution using the Shapiro-Wilk test. Since the groups were normally distributed, a Two-way ANOVA and Tukey’s test were performed to compare the fracture resistance in groups at the significance level of p < 0.05.

Results

The fracture resistance and statistical comparison for each group were demonstrated in Table 2. The highest fracture resistance was recorded in Group III, while the lowest fracture resistance was shown in Group VI. No statistically significant difference was found between the immediate groups (Group I, II, III) (p > 0.05). Additionally, no statistical difference was found between the delayed groups (Groups IV, V, VI), (p > 0.05). When comparing the restorations that were made immediately or procedures delayed for 3 months, no statistical difference was found between all groups (p = 0.497) (Table 2).

Table 2 Mean and standard deviation of fracture resistance for each experimental group.

	Mean	Std. deviation	Std. error	Minimum	Maximum	p	df	
Group I	942.40a	217.30	68.71	734.80	1,366.09	0.497	5	
Group II	884.29a	313.46	99.12	402.20	1,343.93	
Group III	1,002.81a	222.22	70.27	494.96	1,308.82	
Group IV	814.38a	290.37	91.82	540.40	1,265.18	
Group V	963.71a	376.01	118.90	415.24	1,357.27	
Group VI	793.37a	241.29	76.30	556.19	1,350.78	
Note:

a : No difference between with the same letter. (Two-way Anova with 5% as level of significance and Tukey test).

According to the FE-SEM views (Fig. 2), while the most unrepairable fractures were obtained in Group VI, the most repairable fractures were obtained in Group I (Table 3). Figures 2A, 2B: A repairable fracture line was observed above the CEJ. Figures 2C, 2D: The fracture line in both the restorative material and the tooth extends below the CEJ. It is an example of an irreparable fracture type. Figures 2E, 2F: The fracture line starting from the crown and extending to the root was observed.

Table 3 Percentages of fracture modes in experimental groups.

Groups	Repairable (%)	Unrepairable (%)	
Group I	9 (90)	1 (10)	
Group II	8 (80)	2 (20)	
Group III	6 (60)	4 (40)	
Group IV	3 (30)	7 (70)	
Group V	1 (10)	9 (90)	
Group VI	0 (0)	10 (100)	

Discussion

ETT are less resistant to occlusal forces than intact teeth (Sedgley & Messer, 1992; Mickevičienė, Lodienė & Venskutonis, 2020). Starting from this point, restoring these teeth aims not only to rebuild tooth morphology but also to provide strength. The aim of this study was to compare the fracture resistance of restorations that were restored immediately and delayed for 3 months after RCT.

It is known that posterior teeth are exposed to more chewing forces and, as a result, weaker than anterior teeth (Hansen, Asmussen & Christiansen, 1990; Sedgley & Messer, 1992). The amount of tooth structure after RCT plays a role in determining the treatment technique and choice of the material for the restoration (Hansen, Asmussen & Christiansen, 1990). When one marginal ridge was lost, tooth rigidity decreased by 46%, and when a MOD preparation was made, relative cuspal rigidity decreased by an average of 63% (Hansen, Asmussen & Christiansen, 1990). Although indirect restorations such as inlay, overlay, and crown restoration after RCT have been popular recently, especially in large MOD cavities, improved direct restoration techniques are commonly preferred in Class II DO or MO cavities (Donly et al., 1999; Ferracane, 2006). For that reason, composite resin, bulk-fil composite resin, and polyethylene fibers were used for the restoration of root canal-treated maxillary premolar teeth with Class II DO cavities in the present study.

In some of the studies, a compressive force was applied to the specimens using a universal testing machine, and a variety of metallic load devices such as steel spheres, steel cylinders, and wedge-shaped objects was used (Donly et al., 1999; Soares et al., 2006; Plotino et al., 2008). It was claimed that the most effective way to determine the fracture resistance of premolars is to use a metallic ball with a specific diameter (Burke, Wilson & Watts, 1993). From this point, a 3 mm diameter stainless-steel sphere was chosen for vertical compressive loading in this study.

This study aimed to evaluate the effect of delayed permanent restorations on the fracture resistance of the ETT premolar teeth. It was pointed out that ETT should be restored with final restoration as soon as possible to improve the result and prognosis of RCT (Donly et al., 1999). Mindiola et al. (2006) reported that restoration should be completed until 90 days. In this study, taking into account the conditions under which temporary fillings had to be placed. The permanent restorations were delayed for 3 months. Thermo-mechanical aging procedures were preferred to simulate the oral conditions for a 3-month delay.

Direct restorations have become easier and safer in ETT in recent years with developments in restorative materials. Bulk-fill composite resins enable the placement of up to 4 mm of material in a single step (El-Damanhoury & Platt, 2014; Al Sunbul, Silikas & Watts, 2016). Additionally, these materials reduce chair-side time and simplify clinical processes (El-Damanhoury & Platt, 2014). The other newly presented material is polyethylene fiber technology. Ribbond is a leno-woven, ultra-high-molecular-weight polyethylene fiber with an ultrahigh elastic modulus (Lassila, Nohrström & Vallittu, 2002). Tekçe et al. (2016) suggested using polyethylene fiber under the flowable bulk-fill composite resin. Belli, Dönmez & Eskitaşcioğlu (2006) reported that using Ribbond under the hybrid composite resin enhanced the fracture strength of mandibular premolar teeth with MOD cavities. Hshad et al. (2017) found that the application of Ribbond under the hybrid composite increased the fracture resistance of endodontically treated mandibular teeth. In the present study, although the highest fracture resistance was found in the group with fiber insertion under flowable bulk-fill composite immediately after RCT, no statistically significant difference was obtained (p > 0.05). Hypothesis 1 was, therefore, accepted. This result in our study may be caused by the application of flowable bulk-fill while hybrid composite resin was used in the other study. Estelite Bulk-Fill Flow was developed with radical amplified photopolymerization technology and suprananospherical filler (Tokuyama, 2018). Additionally, while MOD cavity in mandibular premolar teeth were prepared in the other studies (Belli, Dönmez & Eskitaşcioğlu, 2006), class II cavity in maxillary premolars were prepared in our study. These advanced characteristics could be reason why the results differ from earlier research.

Similar to the results of our study, Shafiei et al. found no significant difference between flowable bulk fill with or without Ribbond in endodontically treated maxillary premolars (Shafiei et al., 2014). Recently, Eliguzeloglu Dalkılıç et al. (2019) compared the samples that were intact/subjected to thermomechanical aging and restored by flowable bulk fill composite (Tokuyama Estelite Bulk-fill, Tokuyama Dental, Tokyo, Japan) with or without Ribbond, and they reported that there was no significant difference among the groups.

One of the main goals of this study was to search for the effect of applying permanent restoration with a 3-month delay on the fracture resistance of the ETT premolar teeth. According to the results, there was no statistically significant effect on the fracture resistance of the teeth between the restorations made immediately and after a 3-month delay (p > 0.05). Therefore, the second null hypothesis was accepted.

Although there was no statistically significant difference between the groups in this study, maximum fracture resistance was achieved in the groups, which was restored immediately by inserting Ribbond under the flowable bulk-fill. Teeth restored with Ribbond under the flowable bulk-fill composite resin after a 3-month delay showed the lowest fracture resistance. The lowest fracture values could be related to the inadequately removed temporary filling material from the cavity, which may negatively affect the bonding between teeth and the restorative material. Besides this, temporary materials are exposed to occlusal forces, temperature changes, and dimensional changes (Sedgley & Messer, 1992). All these factors may affect the adhesion of materials and fracture strength of teeth after RCT.

The repairable property of composite materials provides some advantages (saves time and cost) for clinicians and patients. In this study, it was determined that most of the fracture types in restorations performed immediately after RCT were repairable, while most of the fracture types in delayed restorations were irreparable.

This in vitro study had some limitations. The fracture load was conducted under 50 N continuous direction and speed. However, to mimic the oral environment, the type and the direction of load were crucial and effective in reducing tooth fracture resistance. Besides, the biomechanical properties of periodontium were not simulated. The long-term prognosis for extensive direct composite resin restorations of endodontically treated premolars requires more clinical research.

Conclusion

Within limitations, the following were concluded: 1) The application of polyethylene fiber increased the fracture resistance in immediate restorations of endodontically treated maxillary premolars.

2) Nano-hybrid composites are also preferred, as well as bulk-fill composites in endodontically treated maxillary premolars with Class II DO cavity preparations.

3) Most of the unrepairable failure modes were detected in the group which was restored by inserting Ribbond under the flowable bulk-fill composites after 3-month delay.

4) Teeth were restored immediately after RCT showed more repairable fractures.

Supplemental Information

Supplemental Information 1 Fracture resistance values of each group.

Additional Information and Declarations

Competing Interests

The authors declare that they have no competing interests.

Author Contributions

Zeynep Buket Kaynar conceived and designed the experiments, performed the experiments, prepared figures and/or tables, and approved the final draft.

Gözde Akbal Dinçer conceived and designed the experiments, performed the experiments, authored or reviewed drafts of the article, and approved the final draft.

Nazmiye Donmez analyzed the data, authored or reviewed drafts of the article, and approved the final draft.

Human Ethics

The following information was supplied relating to ethical approvals (i.e., approving body and any reference numbers):

This study was approved by Bezmialem Vakif University Ethics Commitee (E-54022451-050.05.04-89852).

Data Availability

The following information was supplied regarding data availability:

The raw measurements are available in the Supplemental File.

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
