# Peer review of "Comparison of fracture resistance between immediate and delayed composite restorations with or without fiber after root canal treatment: a field-emission-gun scanning electron microscope study"

_PeerJ, doi:10.7717/peerj.19018_

## Round 0.1 · original submission · Major Revisions

Dear authors,
Based on the reviewers' feedback, the manuscript requires major revisions. Both reviewers have highlighted the need for corrections in grammar and terminology, particularly in the abstract and throughout the text. Additionally, the null hypothesis is incorrectly framed and must be corrected. Reviewer 2 has provided detailed comments on the methodology, results, and tables, including the need for additional references, clarification of experimental details, and more robust statistical analysis. Please address these concerns thoroughly to improve the manuscript's clarity, accuracy, and overall quality before resubmission.

·

Basic reporting

Grammar and English needs minor corrections at some places.
In the abstract section it should be affect instead of effect.
In the discussion section 3 is repeated in 3months (3 3 months)

Experimental design

Null hypothesis is when there is no statistically significant difference between the groups. In the manuscript actually alternate hypothesis is written stating it as null hypothesis and then it is rejected. Kindly correct that part.

Validity of the findings

No Comment

Additional comments

Well thought out and conducted research. Needs some minor changes.

Reviewer 2 ·

Basic reporting

First, i want to thank the authors. There are few comments that i want to mention here.

In abstract
Please correct the mistakes in line 8 (aftersubjected) and line 9 (andexposed)
What do mean by with temporary restoration in line 10?
Line 12, SEM should be mentioned first fully.
Lines 15, 16, 17, please write the conclusion in a better understandable way.

Introduction
Sufficient field background/context was provided with good literature
The study is self-contained with relevant results to hypotheses

In tables and figures
Table 1, please batch number or lot number
Table 2 add df and One way anova above the p value
Add footnotes for the abbreviation , what is the meaning of a?

Experimental design

Original primary research is within Aims and Scope of the journal. Research question is well defined, and relevant .

In methodology
Line 70, Add the inclusion and exclusion criteria of the selected teeth
Line 76 add reference
Line 78 K file (brand and country)
Line 80 and 81, add NaOCl (brand and country) and correct this %2.5 NaOCl
Line 86 please mention that the delayed were restored with temporary filling and mention the type of T. restoration.
Line 104 add reference
Line 107 add reference
Line 107 did the authors examine the teeth for possible cracks or hidden fractures after the 3 months of thermo-mechanical. How was the temporary restoration removed ? Did the authors examine the tooth after temporary restoration removal? Cracks can occur and lead to unrepairable fracture. So in my opinion the comparsion among the delayed group is not valid and bias as no asesstment was done in the teeth after the thermomechanical aging.
Line 114 rephrase please
Line 124 SEM should be mentioned fully.
Line 125 mention the brand and country of the Universal testing machine
Line 132 please elaborate more on the mode of FESEM (SE) and KV and magnification which were used
Line 134 correct referencing style (17)
Line 136 mention the name of the statistic software
Line 135 Two-way Anova may be more suitable for your study. Can you please justify your choice of one-way Anova?

Validity of the findings

Results
Line 143 to 147, the results are not properly written. No multiple comparison was made among the groups and between them. One-way Anova, only gives you overall outcome of the analysis. The authors are needed to do multiple group comparison using Post hoc tests. Need to mention df value in table 2. Please mention One-way Anova term in the table as well.
Line 148, more elaboration is needed about the results from FESEM.
Line 154-155 correct the Grammer
Line 169-170 incomplete sentence
Line 203-211 mistakes present
Line 215, how inadequate is it ? what was the type of restoration? did one operator examine the teeth or two operators?
Line 220 correct the grammar
Line 227 mention the authors recommendation please
Line 231 Mention Class II DO

Additional comments

References section
Line 240 References need to be corrected, Title and journal name mistakes were detected.

---

## Round 0.2 · Minor Revisions

Dear authors,

Thank you for submitting the revised manuscript. After reviewing the changes, one of the reviewers recommends a minor revision to address further issues.

·

Basic reporting

In the abstract section, please correct the spelling of difference in the result section.

Experimental design

No comment

Validity of the findings

No comment

Additional comments

No comment

Reviewer 2 ·

Basic reporting

Thank you for submitting the revised version. Here are my comments.
Abstract:
1. Please add (groups)
'While ________ I (Composite resin), II (Flowable bulk-ûll) and III (Ribbond+Flowable bulk-ûll) were restored immediately'
2. Row 8 in the abstract, please add distilled water.
3. Correct the word difference in row 14 of the abstract.

Title:
1. Your title is misleading. Do you need to mention the polyethylene only while you are having 6 different groups?
2. Please mention immediate before delayed.

Introduction
1. Row 18 and 19: Reorder this sentence, please. It should be: preparation, irrigation and obturation procedures and the quality…...
2. Row 53, add year to the reference.

Experimental design

Methodology
1. Row 70: What do you refer to by "non-occlusion"? Please correct.
2. Row 86: What is the method used in the random allocation of teeth into two groups?
3. Row 119: After thermo-mechanical aging, please mention the methods used in the removal of the temporary restoration.

Validity of the findings

Results:
Row 152: Add immediate before groups
Row 152: Please delete this sentence "which made permanent restorations immediately after RCT"
Row 153: Add the p value.
Row 153: Add delayed restoration before groups
Row 154: Delete this "which made 3-month delayed permanent restorations"
Row 158-162: rewrite this paragraph properly please "Figure 2 (a,b): A fracture line was observed above the cementoenamel junction at a repairable. Figure 2 (c,d): The fracture line in both the restorative material and the tooth extends below the cementoenamel junction. It is an example of an irreparable fracture type. Figure 2 (e,f): The fracture line starting from the crown and extending to the root was observed."

Discussion
Row 226: Null hypothesis was accepted. correct, please

Conclusion:
Row 249: of endodontically instead of after endodontic

Table 2: Please rewrite what is mentioned in a
You can utilize the footnotes in a better way.

Additional comments

None

---

## Round 0.3 · accepted · Accept

Dear authors,

After careful review and consideration, we are pleased to inform you that your manuscript has been accepted for publication. We appreciate the effort and dedication you have invested in your research, and we are confident that your work will make a valuable contribution to the field.

Reviewer 2 ·

Basic reporting

Clear and unambiguous, professional English used throughout

Row 1: A suggestion for a better title is: Comparison of fracture resistance between immediate and delayed composite restorations with or without fiber after root canal treatment: a field-emission-gun scanning electron microscope study
Row 109: Please add "and cavities were restored with flowable bulk fill composite resin."

All the best and best regards

Experimental design

Rigorous investigation performed to a high technical and ethical standard

Validity of the findings

All underlying data have been provided; they are robust, statistically sound, & controlled.

Additional comments

No more comments